# Machine Learning Algorithms for Activity-Intensity Recognition Using Accelerometer Data

**DOI:** 10.3390/s21041214

**Published:** 2021-02-09

**Authors:** Eduardo Gomes, Luciano Bertini, Wagner Rangel Campos, Ana Paula Sobral, Izabela Mocaiber, Alessandro Copetti

**Affiliations:** 1Computer Science Departament, Fluminense Federal University, Rio das Ostras 28895-532, Brazil; eduardogomes@id.uff.br (E.G.); lbertini@id.uff.br (L.B.); wrcampos@id.uff.br (W.R.C.); 2Production Engineering Departament, Fluminense Federal University, Rio das Ostras 28895-532, Brazil; ana_sobral@vm.uff.br; 3Natural Sciences Departament, Fluminense Federal University, Rio das Ostras 28895-532, Brazil; imocaiber@id.uff.br

**Keywords:** pervasive healthcare monitoring, activity and intensity recognition, mobile computing, machine learning, accelerometers

## Abstract

In pervasive healthcare monitoring, activity recognition is critical information for adequate management of the patient. Despite the great number of studies on this topic, a contextually relevant parameter that has received less attention is intensity recognition. In the present study, we investigated the potential advantage of coupling activity and intensity, namely, Activity-Intensity, in accelerometer data to improve the description of daily activities of individuals. We further tested two alternatives for supervised classification. In the first alternative, the activity and intensity are inferred together by applying a single classifier algorithm. In the other alternative, the activity and intensity are classified separately. In both cases, the algorithms used for classification are k-Nearest Neighbors (KNN), Support Vector Machine (SVM), and Random Forest (RF). The results showed the viability of the classification with good accuracy for Activity-Intensity recognition. The best approach was KNN implemented in the single classifier alternative, which resulted in 79% of accuracy. Using two classifiers, the result was 97% accuracy for activity recognition (Random Forest), and 80% for intensity recognition (KNN), which resulted in 78% for activity-intensity coupled. These findings have potential applications to improve the contextualized evaluation of movement by health professionals in the form of a decision system with expert rules.

## 1. Introduction

The advancement of wireless technologies allowed the use of sensor devices in environments such as homes and clinics, as well as in people’s own bodies, enhancing pervasive computing and wearable computing. This is making it possible to design new mechanisms for remote patient monitoring, and to improve the systems analysis and accuracy. In health monitoring, an essential requirement in the person daily monitoring is the recognition of activities such as sleeping, sitting, walking, or performing an activity of daily living (ADL). One way to achieve this recognition is with the support of data provided by 2 accelerometer-type sensors [1,2,3].

Although day-to-day activities are apparently situations of simple recognition, developing an algorithm using as few sensors as possible, so that the system does not bother the user, is not so simple. Reference [4], for example, uses five sensors distributed throughout the body, which can compromise the user’s freedom of movement. In addition, there is a greater challenge in this work, which is our final objective, in the joint detection of the activity with the intensity of the movement.

Activity recognition monitoring systems are essential tools for health professionals. Such systems can ensure early intervention and are useful in rehabilitation and prevention. We can imagine activity recognition systems that analyze information on the number of times and the instants at which a patient experiences changes in his or her physical routine. A practical application can be the identification and analysis of the moments when the patient exhibits slowness of movement (bradykinesia). Vigorous movements, for example, excessive tremors or shaking [5] can also be identified.

Some papers [6,7,8,9] have already demonstrated techniques for recognizing human activities, especially with the use of machine learning algorithms. Recently, some researchers have targeted specific health contexts, such as diabetes [10], and hypertension [11]. This diversity in the areas of application in health requires greater interaction between the knowledge of the specialist and the designer of the system. Therefore, there is a need for models that can express concepts and rules, and facilitate the interpretation by the expert.

One aspect that has not been addressed in works that involve the recognition of activities is the recognition of the intensity level of the activity. Generally, the interpretation given for the intensity is concerning the effort that the activity demands. In this work, we emphasize the need to consider the movements intensities that are performed during an activity. In scientific research in the area of health, the intensity is stratified in terms of light, moderate, or vigorous activity [12]. In this sense, it is important to consider this parameter in the recognition of activities. In the biomedical area, the intensity of movements has been determined using specific physiological methods that involve the calculation of energy expenditure [13]. However, in certain approaches, it is important to have information regarding the intensity of the movement through simpler methods and without the main interest being the energy expenditure.

The coupling of activity with intensity, which we named activity-intensity, brings the possibility of better representing the knowledge of health professionals in the form of decision systems with specialist rules. Examples of rules in a decision system, as presented in Reference [14], can be established as follows: (a) if the type of activity is domestic, the intensity is vigorous, and the heart rate is high, the patient can be considered as alert (and not in a medical emergency). Thus, changes in vital signs influenced by changes in behavior could be tolerated based on rules that take these factors into account; (b) if the activity is sleeping and the intensity is vigorous, the possibility of insomnia increases. The two examples show different intensities for each activity.

The present work focuses on the role of the intensity discrimination in the recognition of distinct activities. Thus, we propose the Activity-Intensity coupling to improve the mechanisms of activity recognition. Additionally, we evaluate the best technique to perform the recognition using supervised machine learning.

## 2. Materials and Methods

### 2.1. Recognition of Activities Coupled with Intensity

The intensity of an activity, from the perspective of the physiological effort expended, is usually determined by the Metabolic Energy Expenditure or MET. The MET [12] reflects the energy consumption associated with performing an activity. The MET is, by definition, the ratio of the metabolic rate during physical work to the standard basal metabolic rate of 1.0. One unit of MET is considered the resting metabolic rate while sitting still. Then, the following intensity levels can be established systematically: light (less than 3.0 METs), moderate (3 to 6 METs), and vigorous (more than 6 METs) [12]. Intensity levels are assigned to each activity as a function of the energy expenditure expressed in METs. This effort classification depends on the type of activity and the exercise intensity degree, generating a representative value (e.g., cycling: 8.0 METs and meditating: 1.0 MET). Thus, the movement intensity is usually assigned a relative interpretation of the energy expenditure [12].

Physical activity is defined as any bodily movement performed by the musculoskeletal structure that leads to an additional expenditure of energy than consumed during rest [15]. Thus, the amount of energy expended in the execution of the movement is the criterion used to differentiate between physical activities. The measurement of energy expenditure is widely discussed in terms of consumption of oxygen and heart rate, and accelerometer has been discussed [16,17]. Despite the complexity of the measurement of physical activity, accessible and inexpensive instruments are of interest to researchers and clinicians. In this sense, the use of motion sensors appears to be interesting and suitable are based on the idea that body movements and acceleration parameters are a reflection of the energy expenditure [16].

In Reference [13] the author investigated better approaches for activity recognition to define an energetic expenditure value of the activity. Some recommendations were given such as the position of the accelerometer, which should be standardized, with the most common places being the lower back or the waist (probable center of mass of the body). Another point considered by the authors is that when accelerometer output is expressed as counts (or vector magnitude), the problem arises that the relationship between counts and energy expenditure will differ depending on the type of activity performed. Activity recognition may provide a solution to this problem because a different value of energy expenditure can be assigned for each activity/intensity combination.

Some studies in the area of Pervasive Computing and the Internet of Medical Things follow the interpretation of effort for intensity. References [18,19] proposed a system to recognize activities and measure their intensities. An experiment was developed with three accelerometers positioned in the thigh, chest, and waist. The activities analyzed included running, climbing and descending stairs, walking, standing, sitting, and laying down. The hit rate was approximately 82%. The heart rate was used to measure the intensity; however, the treatment and analysis of these data for intensity rating are not discussed.

In the study by Reiss [20], variations in Adaptive Boosting, which consists of building a stronger classifier from a series of weaker classifiers, techniques to estimate the intensity of activity as one of the three categories (light, moderate, and vigorous) were used. Using the database PAMAP2 [21], which has activity data monitored with three accelerometers and a heart rate monitor, results close to 90% were obtained to estimate the intensity. For example, activities such as sitting, working on the computer, and watching TV were classified as light intensity.

In Reference [6], a framework is proposed for the recognition of 19 activities with different levels of intensity, such as domestic tasks, locomotion, video games, exercises, and playing sports. The intensities are divided into sedentary behavior, light activity, moderate activity, and vigorous activity. The data was obtained from an ActiGraph GT3X tri-axial accelerometer positioned on the right side of each participant’s waist. In the pre-processing phase, a proprietary filter is applied and the data was transformed to the frequency domain. Considering only continuous movements, an average F-Score of 98% is reached, whereas the best results for intermittent movements is 84%. The best F-Score for the recognition of all activities together is 64%.

A problem associated with activity recognition systems that use multiple sensors in the body is that they are not practical to be used in daily life. For example, in References [19,20], three accelerometers were attached to the body which makes the system unsuitable for regular use. To overcome this problem, in the present study, we propose the utilization of a single smartphone accelerometer positioned in the waist, which is the typical location to carry a smartphone. Although it is possible to collect physiological data in real-time with smartphones, the objective of this work is to provide a system that avoids contact between the device and the patient’s body. Thus we opted for only the accelerometer, as it is directly related to physical movement and has already been proven effective [22].

According to Reference [4], the activity considered contains only repetitive weight exercises (six upper body strength training exercises), using five sensors positioned on the arms and chest. The intensity is associated with the muscular strength used in each exercise sequence, using different weight loads. After each series of repetitions, participants were asked to self-assess the intensity of the exercise according to a standard scale. The article considers activity and intensity, but the intensity is interpreted differently from the interpretation that we give in this work, being associated with the strength necessary to execute the movement, and not in the amount of movement.

In contrast to other studies, in our current approach, we consider the intensity as a critical information for the recognition of each activity. For example, walking has an intrinsic intensity, which is different from the intensity associated with the activity of laying down. Thus, the subdivisions of intensity must consider its inherent activity, instead of using it as a general rule of evaluation, such as other approaches [6]. We also adopt the same classification of intensity, but without associating it with energy expenditure.

In this work, we investigated three activities considered basic for a person at home: laying down, sitting, and walking. While the first two activities are postures, walking is an ambulation activity, which requires continuous movement. We propose the activity-intensity approach in the recognition of the referred basic activities. An example of the coupling of activity and intensity, even when the person is laying down, is the execution of occasional movements at different intensities.

Each activity is evaluated at three levels of intensity: light, moderate, and vigorous (9 situations). Each situation corresponds to the following examples of everyday life: light laying down—the whole body static on the bed; moderate laying down—at certain times, the body is turned to change position; vigorous laying down—a simulation of chronic insomnia where all the limbs, including the head, move; light sitting—fully relaxed lower limbs in the seat; moderate sitting —restless legs and arms practically relaxed; vigorous sitting—upper and lower limbs active; light walking—slow walking; moderate walking—normal walking; vigorous walking—walking with quick steps.

### 2.2. Data Collection

An experiment was conducted with 8 healthy participants (6 men), in the 15–50 years old range. The project was approved by the local ethics committee of Federal Fluminense University, Rio de Janeiro, Brazil, and each participant gave written informed consent before participation. For the calculation of this sample size, the methodology described in [23] was used, resulting in 8 participants as a function of the small standard deviation in the accelerometer data produced by the device. Other studies have also presented similar experiments with few participants. Reference [4] assessed how the number of participants impacts the classification of physical exercises, showing that the error of the intensity prediction was stable after adding the sixth participant.

Each participant received a smartphone placed inside a waistband, as shown in Figure 1. The participants were instructed to naturally execute each movement condition without receiving any stimulus or corrections during it, all situations were performed inside our laboratory, without a pre-defined route. The smartphone accelerometer generated the gravity acceleration data at a rate of 20 Hz, with a range of ±2 g, and the 3 axes information were stored on the smartphone together with the timestamp. The smartphone chosen was a 1 GHz processor, 1 GB RAM, Android OS 4.1.2, i.e., with severe resource limitations.

Data collection for each combination of activity and intensity was initiated when the participant pressed a start button. This interaction did not require the smartphone to be removed from the waistband. After 1 min of data collection, a signaling beep was emitted. In the lying posture, the participant used a mat, and in the sitting posture, a chair. In the walking activity, mainly in moderate and vigorous intensities, the participant left the room and walked a 20 m corridor. The total experiment time for each participant was an average of 15 min, which included a training phase of each behavior scenario and the execution of the 9 specific behaviors of 1 min each, with intervals of 30 s. A total of 10,000 accelerometer raw data was collected for each situation. After the data collection and storage, the preprocessing phase was conducted.

### 2.3. Preprocessing and Extracting Features

In the preprocessing phase, the data are prepared to optimize the classification performed by machine learning algorithms. In the present work, we divided this step into three parts:(a)Removal of corrupted data and outliers. Firstly, all corrupted data (including missing data) returned by the smartphone were removed. Because the data are multivariate, we apply the technique of hierarchical clustering to remove outliers. Hierarchical clustering is an unsupervised technique that calculates the distance between each point present in the database, clustering them according to proximity. To define outliers using this technique, we need to analyze which clusters are divergent from the others based on a threshold of the Euclidean distance. This threshold is drawn in Figure 2 as a horizontal line. We can observe in this figure that clusters 8 and 21 are connected to other ones at much higher distances. Thus we can define both clusters as outliers.(b)Segmentation and application of statistical metrics. The data were segmented by a window of 2.5 s, without overlap. The following statistical metrics were calculated for each axis (x, y, and z) present in each window: mean and standard deviation. In total, the feature vector had 6 features. After the segmentation process, the dataset contained 2000 samples. Metrics were calculated to optimize the data window and to minimize the impact of univariate outliers.(c)Analysis of distribution and standardization of data. In this step, we verify the normality of the data. The analysis showed that our data is not normally distributed. The algorithms used in this experiment do not require normal data to perform well.
(1)$$x_{new}=\frac{x-\mu }{\sigma }$$The data standardization was performed by applying the Equation (Equation 1), where *x* is the sample, μ is the mean, and σ is the standard deviation. The standardization was performed to match the variables so that all axes had the same weight in the analysis, regardless of their initial amplitude. Without standardization, the algorithms for recognition may assign greater importance to variables with larger amplitudes.

### 2.4. Classification

We defined two possible "classification alternatives" for Activity-Intensity recognition. The first alternative is to classify the data directly using only one classifier. In this case, we will have a classifier that infers from the 9 possible classes, for example, light walking. In the second alternative, a first classifier infers the activity, for example, walking, and a second classifier infers the intensity, for example, light. The second alternative would be important to mitigate the inter-class variability, since we will first recognize the activity, probably with high accuracy, and then the intensity, which classification is more difficult.

Based on the data analysis performed in Section 2.3 and the literature, we chose three algorithms to use in the recognition: K-Nearest Neighbors (KNN), Random Forest (RF), and Support Vector Machine (SVM). To find the best hyperparameters for our experiment we applied the Grid Search algorithm. Grid Search builds a model for every specified combination of hyperparameters and returns the accuracy of each model. All implementations were made using the Python library scikit-learn.

In the test phase, for each of the three mentioned algorithms, the samples were divided into training and test subsets using the k-fold cross-validation method with k=10. In the k-fold cross-validation, the learning set is partitioned into *k* disjoint subsets of approximately equal size. This partitioning is performed by randomly sampling cases from the learning set without replacement. The model is trained using k−1 subsets, which, together, represent the training set. Then, the model is applied to the remaining subset, which is denoted as the validation set, and the performance is measured. This procedure is repeated until each of the *k* subsets has served as the validation set. The average of the *k* performance measurements on the *k* validation sets is the cross-validated performance [24].

## 3. Results

In this section, we present the results of supervised learning methods for classification to investigate the most appropriate technique for activity recognition. The main objective of the experiments was to investigate if it is possible to obtain good accuracy and generalization in the recognition of Activity-Intensity for certain activities.

Table 1 shows the results for the first alternative, which uses only one classifier. We can see that KNN and Random Forest obtained better results with 79% and 77% accuracy, respectively. For this reason, these two algorithms were chosen to be compared in more depth later. This same observation can also be made from the value of the F-Score, which takes into account both accuracy and recall.

A more detailed observation of the results shown in Table 1, which highlights the two better algorithms, can be made from Figure 3 and Figure 4, analyzing which activities are more complex, to understand the intra-class and inter-class variability. These two figures show the confusion matrices of the two algorithms for the first alternative. We can see that KNN and RF had difficulty in intra-class variability since most of the incorrect classifications were inside the same class. Regarding inter-class variability, both classifiers had a satisfactory result with 45 misclassifications for KNN and 40 for RF. The difficulty of obtaining high accuracy in the recognition of activities walking and sitting with the smartphone on the waist is corroborated in the literature, for example in Reference [25].

Regarding the low accuracy of walking activity observed in Table 1, also shown in Table 2, we suspect that it is caused by the participant’s physical parameters variabilities, such as height, weight, and even the interpretation given by the participant to the fuzzy terms inherent to the intensity. The impact of these personal parameters was not the focus of this research and should be investigated better.

Table 2 shows the accuracy obtained with each algorithm for each of the situations (activity-intensity). The activity with less accuracy for both classifiers was walking moderate with 54% accuracy. For the second alternative, which consists of two steps, Table 3 displays the activity recognition results and Table 4 displays the results for the recognition of the intensity of each activity. The best algorithm for this alternative was RF, which obtained 97% for all used metrics in activity classification and intensity recognition.

In Table 3 and Table 4 we conclude that both alternatives obtained similar results for all performance measures evaluated, since it obtained 97% accuracy for activity and 80% for intensity (best scenarios), totaling an accuracy of 78% for the recognition of activity-intensity. In this alternative, when obtaining the activity with the first classifier (e.g., walking), the second classifier will only define the intensity of the activity (light, moderate, and vigorous) already defined. This probably improves the efficiency of the second stage of classification.

Using the second alternative, with two classifiers placed in sequence, the combination of algorithms with the best results were RF (activity) + KNN (intensity), with 78% of total accuracy. However, the recognition of activities is performed through an application on a smartphone, which has a shortage of resources, especially the battery. Therefore, the chosen classifier must have the lowest cost in terms of consumption of device resources, and with the greatest accuracy.

In order to choose the best classifier among those selected in the experiments (KNN—alternative 1, and KNN + RF—alternative 2), we consider other variables besides simply the accuracy. We used the AHP (Analytic Hierarchy Process) multicriteria decision-making method proposed by Saaty [26], which is widely used in the literature to aid decision making [27,28], including choosing the best algorithm for a given problem [29,30].

The proposed criteria are mean accuracy, time, and memory usage for recognition. Table 5 displays the performance of the classifiers for each criterion.

The next step is to create the comparison matrix. Thus the matrix is defined based on the expert’s knowledge, comparing each attribute according to the importance. For example, in matrix *A* (column 1: Time, column 2: Memory, and column 3: Accuracy), the experts defined that the accuracy has 7 times the importance of time. Then, to verify the consistency, it is necessary to multiply the comparison matrix *A* by the priority matrix *B*. The priority matrix *B* contains the importance of each criterion. For example, the values in Equation (Equation 2) show that the most important criterion is accuracy (0.64), followed by memory (0.279), and then by time (0.072).

In the sequence, the matrix *D* is obtained by dividing the matrix *C* by *B*. Then, we can obtain the value of the largest eigenvalue of the matrix *A* ($\lambda_{max}$). Finally, the value of the consistency ratio (CR) can be obtained by dividing consistency index (CI) by the value of the random index (RI) defined by Saaty [26] for different values of n (number of criteria). In the present work, the value of RI is 0.58.
(2)$$A=\left[\begin{array}{ccc} 1 & 0.2 & 0.143\\ 5 & 1 & 0.333\\ 7 & 3 & 1 \end{array}\right],\;B=\left[\begin{array}{c} 0.072\\ 0.279\\ 0.649 \end{array}\right]$$
(3)$$C=A*B=\left[\begin{array}{c} 0.221\\ 0.855\\ 1.99 \end{array}\right]$$
(4)$$D=\left[\begin{array}{ccc} \frac{0.221}{0.072} & \frac{0.855}{0.279} & \frac{1.99}{0.649} \end{array}\right]=\left[\begin{array}{ccc} 3.069 & 3.065 & 3.066 \end{array}\right]$$
(5)$$\lambda_{max}=\frac{3.069+3.065+3.066}{3}=3.067$$
(6)$$CI=\frac{\lambda_{max}-n}{n-1}=0.034$$
(7)$$CR=\frac{CI}{RI}=0.059$$

The value of the CR is less than the empirical limit value of 0.1 proposed by Saaty [26]; therefore, we can conclude that the judgment made in the comparison of the coefficients is consistent. Once the consistency has been verified, we apply the AHP method to find which algorithm and which alternative yields the best evaluation. To calculate the AHP scores, we need to standardize all performance values. We calculate the inverse (1/x) for time and memory because these are inversely proportional to the accuracy. Accuracy is maintained at its original values, as it is already a performance measure of the type "the bigger the better". Then, each value is normalized by the sum of each column. The best algorithm is defined by the weighted average of the normalized performance parameters, using the values of Matrix B as weights. Table 6 shows the result.

Since the KNN algorithm using alternative 1 obtained the highest result, we can conclude that the best alternative for the recognition of activities is alternative 1 using the KNN algorithm.

## 4. Discussion and Conclusions

Our study presents a new approach, which involves integrating the intensity of movements with the activity being executed, that is, the Activity-Intensity approach. We went a step further in comparison to the methodology generally used for the recognition of activities in other studies. To our knowledge, this approach has not been explored so far, as can be contrasted with other studies in this area, such as [1,7], as well as specific works differently interpreting the intensity parameter [4,6,19,20].

We proposed the intensity concept addition in the activities recognition, which is called the Activity-Intensity approach. To verify the feasibility of the proposal, two alternatives for supervised classification were presented to obtain the maximum accuracy: one single classifier and two classifiers separately. The choice of a classifier should not be based solely on accuracy. On mobile devices, for example, we must consider factors such as memory usage and battery consumption. We used the AHP multicriteria decision-making method to consider these hidden factors during the analysis of the results of the algorithms. In conclusion, in this case, we found that the best solution is KNN implemented with a single classifier (alternative 1).

In this work, we investigate the use of machine learning techniques to recognize human activities and postures at different intensities. The coupling of the activity and intensity parameters allowed the recognition of the human activities considered here (walking, sitting, and lying down) in a more precise and complete way. The results obtained 97% accuracy for activity recognition and 78% for activity-intensity coupled (first alternative). The automatic identification of movements/postures is feasible and useful in the context of health care, where the intensity of movements linked to each specific posture or activity has a distinct clinical repercussion. For example, vigorous movements while lying down can be correlated with sleep disorders. At the same time, intense movements during walking are not necessarily indicative of illness. Thus, the automated provision of this information can assist in the decision making of health professionals.

Given the good accuracy of the results, the Activity-Intensity approach represents a valuable option to better describe a patient’s daily activities. In this case, a greater level of detail can be described by the rules. In addition, the results presented here are expected to be of great interest because the activity-intensity recognition is based on accelerometers, which have a very low cost and are easily accessible.

New applications to understand human behavior can be proposed, based on the good results obtained in the activity-intensity recognition shown in this work. Decision-making systems with if-then rules can trigger alerts and highlight certain health situations that deserve to be further investigated. Personalization will be achieved through a person’s behavior used in the system. When there is a routine change, an action can be taken. For example, for an elderly person, any activity that has vigorous intensity at night could be further investigated. When this information is associated with the location where it occurred, we can determine which rooms/areas are not compatible with certain activities-intensity. Diseases such as Alzheimer in the first stages can be better monitored since activity-intensity monitoring adds more information to the person’s daily life.

In future studies, we intend to explore the application of more sensors embedded in a smartphone, such as a gyroscope and a GPS, which will probably improve the information brought by the recognition of types of activities. We also intend to investigate new recognition approaches to improve the accuracy of walking activity and apply techniques such as clustering to group the different levels of intensity without the need for supervised training. As a complementary method, a Fuzzy Expert System will be developed to analyze the daily activities of patients employing the classifier developed in the present work. We will also collect data from a larger sample to evaluate the generalization capacity of the developed technique.

We consider this work highly relevant regarding the innovative analysis (activity-intensity approach), the high performance obtained by the classifier, the practical viability of the sensor (a single accelerometer in smartphone), and the potential implications for healthcare settings.

## Figures and Tables

**Figure 1 sensors-21-01214-f001:**
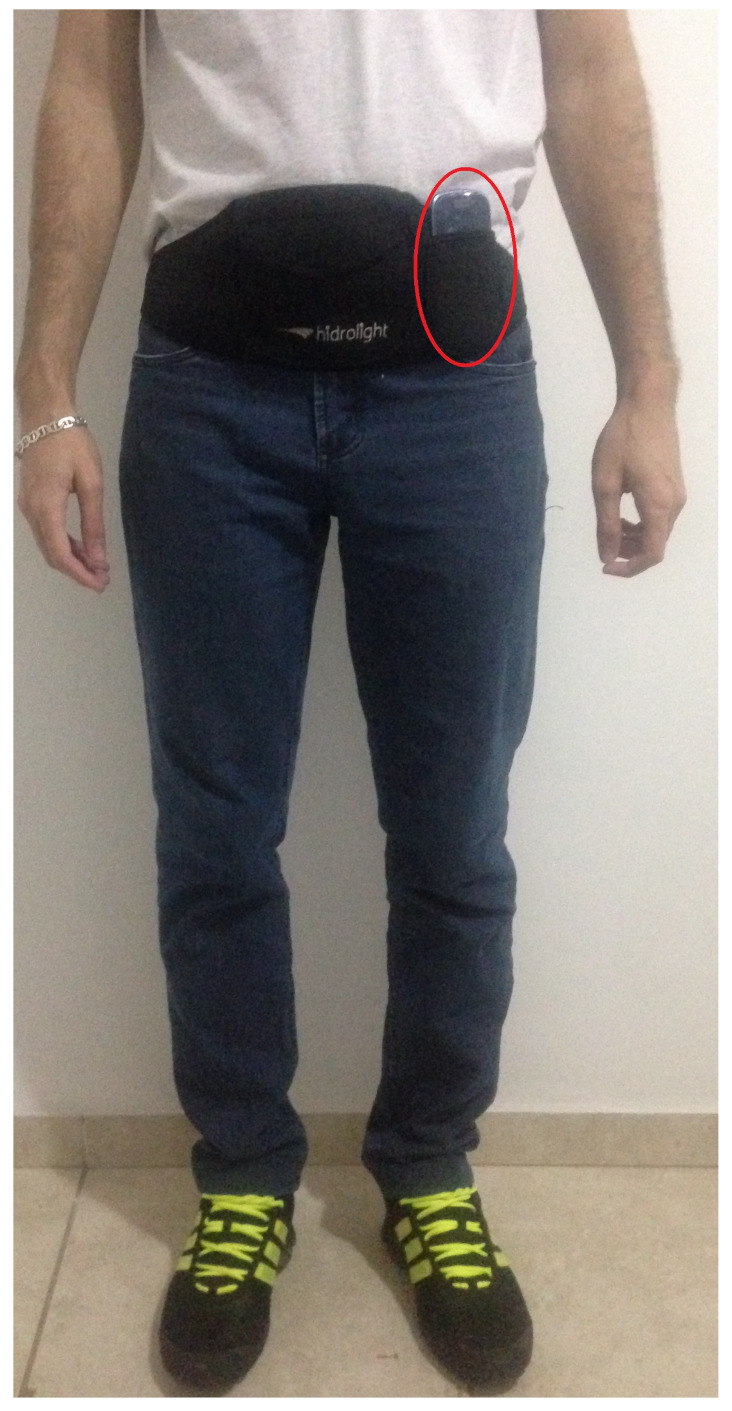
Position of the smartphone on the participant’s body.

**Figure 2 sensors-21-01214-f002:**
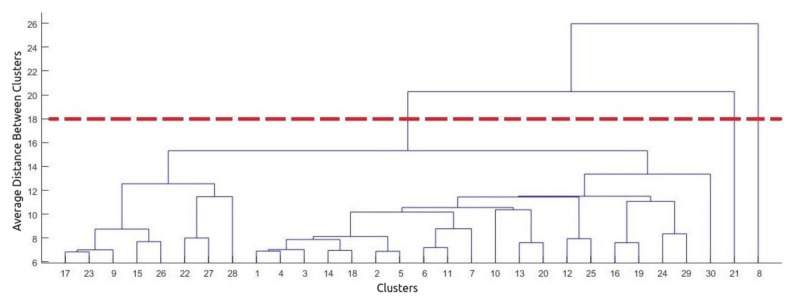
Dendrogram for light walking.

**Figure 3 sensors-21-01214-f003:**
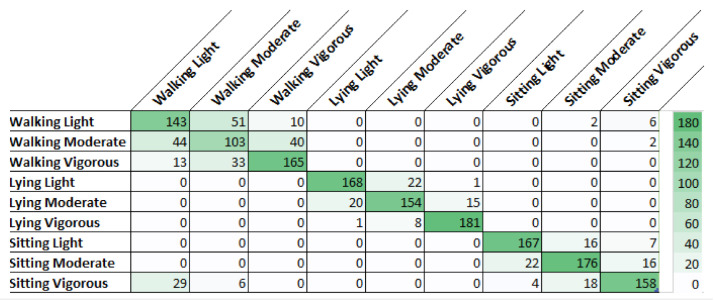
Confusion matrix for classification with K-Nearest Neighbors (KNN).

**Figure 4 sensors-21-01214-f004:**
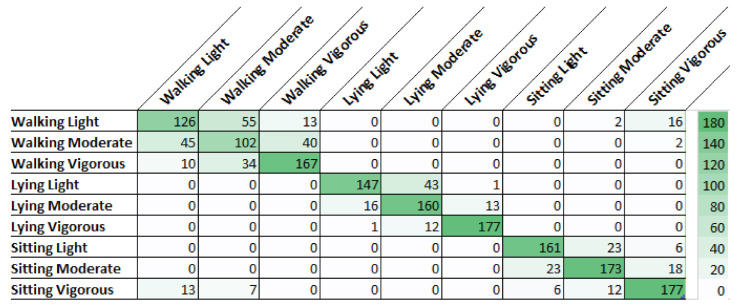
Confusion matrix for classification with Random Forest (RF).

**Table 1 sensors-21-01214-t001:** Alternative 1—Classification with one classifier.

Activity-Intensity				
Algorithm	Accuracy	Precision	Recall	F-Score
KNN	79%	79%	79%	79%
SVM	72%	73%	72%	72%
RF	77%	78%	77%	77%

**Table 2 sensors-21-01214-t002:** KNN vs. Random Forest accuracy—Alternative 1.

Activity-Intensity	KNN	RF
Walking Light	67%	59%
Walking Moderate	54%	54%
Walking Vigorous	78%	77%
Sitting Light	88%	84%
Sitting Moderate	82%	82%
Sitting Vigorous	73%	83%
Lying Light	88%	77%
Lying Moderate	82%	78%
Lying Vigorous	95%	93%

**Table 3 sensors-21-01214-t003:** Alternative 2—Activity recognition.

Activity				
Algorithm	Accuracy	Precision	Recall	F-Score
KNN	96%	96%	96%	96%
RF	97%	97%	97%	97%

**Table 4 sensors-21-01214-t004:** Alternative 2—Intensity recognition.

Intensity				
Algorithm	Accuracy	Precision	Recall	F-Score
KNN	80%	80%	80%	80%
RF	79%	79%	79%	79%

**Table 5 sensors-21-01214-t005:** Performance of classifiers by criterion.

Alternative	Algorithm	Time (s)	Memory (Mb)	Accuracy
1	KNN	0.045	0.145	0.79
2	RF + KNN	0.235	0.637	0.78

**Table 6 sensors-21-01214-t006:** Evaluating the classifier with the alternatives.

Results of the AHP	
KNN—alternative 1	0.80
RF + KNN—alternative 2	0.57

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
