# Peer review of "Machine Learning Algorithms for Activity-Intensity Recognition Using Accelerometer Data"

_sensors, 2021, doi:10.3390/s21041214_

Round 1
Reviewer 1 Report
Please check the comments in the attached file.

Author Response
Reviewer 1:
We would like to thank the editor and reviewers for their insightful comments. We have revised all the text to address the comments received.
- The authors were recommended to revise/discard the following statements, which were meaningless to the research. “Some motivating examples of situations that ilustrates the appropriateness of adopting the activity-intensity approach are:
Ø Excessive movement during sleep: the person, even when laying down, presents different levels of movement that should be identified.
Ø Unusual walking rhythm: walking activity may exhibit characteristics to be explored by experts in the field of walking analysis.
Ø Prolonged sitting on a couch: Analyzing the sitting activity can distinguish moments when the person falls asleep or performs manual tasks. ”
These points have been included to better illustrate the motivation of the research, but indeed they are not that important for the method itself. Then, as suggested by the reviewer, we excluded the cited sentences, without prejudice to the text, since such examples of use in daily life are already mentioned in other parts of the text, in the Introduction and also Conclusion sections.
- Why did authors focus on the activity recognition, which are laying down, sitting and walking. More detailed explanations are required. From my perspective, it is not difficult to recognize the three activities.
We agree with the reviewer that the recognition of only the activity is relatively simple. However, it is shown in the Introduction, as transcribed below, that the objective is the detection of the activity together with the intensity, which makes this recognition more complex. Thus, we believe that the text quoted below, already in the text, is sufficient to elucidate the issue. If the reviewer still finds it necessary to modify the text, please let us know.
“One aspect that has not been addressed in works that involve the recognition of activities is the recognition of the intensity level of the activity” … “In this work, we emphasize the need to consider the intensity of the movements that are performed during an activity”
- More information about the data were required. For instance, which data can be collected from the smartphone, such as heartbeat, blood pressure.
There is a wide range of sensors that can be used on a smartphone, such as for heart rate, positioning, and ambient temperature. In this work we collect data from a smartphone without direct contact with the patient's body, and therefore we cannot collect physiological data in real-time. So we opted for the accelerometer, as it is directly related to physical movement and has already been proven effective. To reinforce this in the text, we added the sentences as follows in subsection 2.1 "Recognition of Activities Coupled with Intensity". We also provided a new citation that addresses the effectiveness of the smartphone accelerometer.
“... To overcome this problem, in the present study, we propose the utilization of a single smartphone accelerometer positioned in the waist, which is the typical location to carry a smartphone. Although it is possible to collect physiological data in real-time with smartphones, the objective of this work is to provide a system that avoids contact between the device and the patient's body. Thus we opted for only the accelerometer, as it is directly related to physical movement and has already been proven effective [22].”
- It is noted that the KNN and RF model did not obtain quite satisfied activitiy recognition considering that the accuracy and precision were lower than 80% (see Table 1).
To answer this question, we added the following paragraph just after Table 1:
“Regarding the low accuracy of walking activity observed in Table 1, also shown in Table 2, we suspect that it is caused by the participants physical parameters variabilities, such as height, weight, and even the interpretation given by the participant to the fuzzy terms inherent to the intensity. The impact of these personal parameters was not the focus of this research and should be investigated better.”
- Please add more explanations about table 2. For instance, I am quite confused that the walking light accuracy was very low.
This comment is addressed in comment #4.
- The following researches are suggested to cite properly: [1] Chen, X., et al., Augmented Ship Tracking Under Occlusion Conditions From Maritime Surveillance Videos. IEEE Access, 2020. 8: p. 42884-42897. [2] M. Mobark, S. Chuprat and T. Mantoro, "Improving the accuracy of complex activities recognition using accelerometer-embedded mobile phone classifiers," 2017 Second International Conference on Informatics and Computing (ICIC), Jayapura, 2017, pp. 1-5, doi: 10.1109/IAC.2017.8280606.
We are grateful for the bibliographical suggestions. The suggested reference [1] is not directly related to our work. The citation [2] is relevant, since the author also had the same difficulty in differentiating the activities walking and sitting using a smartphone on the waist, despite analyzing more complex activities and not treating the intensity. So we decided to cite the work [2] (Mobark et al.) to reinforce the intrinsic difficulty of achieving high accuracy in this type of classification (in Results Section).
Reviewer 2 Report
The authors propose two approaches for recognizing activity and activity intensity using a smartphone accelerometer. Specifically, the authors compare (1) a single classifier activity-intensity classification of 9 classes with (2) a two-step classification of 3-class activities and 3-class intensities using two separate classifiers. In order to determine the best performing approach for activity-intensity recognition, the authors utilize Analytic Hierarchy Process (AHP) that takes into account the time and memory usage of the different approaches along with their recognition accuracy. Although the overall research framework is adequate, many parts of the manuscript lack clear details. I feel that this manuscript must be re-examined thoroughly in its entirety, and in particular, the following parts must be revised.
- The details of the dataset must be given. How many data were used for training and testing? How was the feature vector defined (i.e., size etc.)?
- In section 2.3 (a) the authors describe the process of removing outliers from the collected accelerometer data. Figure 2 shows a dendrogram, but I am not sure what the ‘distance (units)’ and ‘indices’ mean in this context. Does each index signify a data point? Does this mean there are only 30 data points? If the red horizontal dotted line determine the outlier, then based on what? (distance, units?) Do indices #21 and #8 represent the outliers in Figure 2?
- In lines 169-170, the authors state that mean and standard deviation were ‘applied’ (or calculated, perhaps?) to each window. How were these statistics used? It seems that mean and standard deviation were only used for standardizing the data.
- In line 198, the authors observe ‘two’ ‘best’ algorithms (KNN and RF). This expression is strange since ‘best’ usually indicates top ‘one’ of something, not top two.
- I don’t understand what lines 200-201 mean.
- Table 5 is confusing. The authors compare two different approaches: approach using a single classifier vs. approach using two classifiers. However, the alternative 2 in Table 5 lists three methods, KNN, RF, and RF+KNN. Shouldn’t it list only RF+KNN?
- For the two-step approach, the authors report the accuracy to be approximately 80%. I would like to see more detail on how the authors reached this conclusion. Say you have 90% accuracy on activity recognition, and out of the 90% correct data you have 80% correct intensity recognition. In the end, you have 72% of accuracy. In order to reach the conclusion given by the authors, the intensity recognition accuracy given in Table 4 must be completely subsumed by the activity accuracy given in Table 3. Verify if this is the case.
- The AHP description given in lines 234-240 is vague and unclear. What does matrix A and matrix B signify? Where do the numbers in equation (2) come from? Detailed explanation is needed.
- In lines 251-253, the authors stress the novelty of the activity-intensity aspect. Please refer to the following paper: Pernek et al., “Recognizing the intensity of strength training exercises ith wearable sensors,” Journal of Biomedical Informatics, 58, 2015.
(https://www.sciencedirect.com/science/article/pii/S1532046415002142)
- Last but not least, the manuscript should undergo an extensive proofreading by a native English speaker. Some example of typos and errors are listed below:
- Line 23: activity daily living → missing ‘of’ (activity of daily living)
- Line 56: recognition o distinct
- Line 87: depending of the type → ‘on’ instead of ‘of’
Author Response
Reviewer 2:
We would like to thank the editor and reviewers for their insightful comments. We have revised all the text to address the comments received.
The authors propose two approaches for recognizing activity and activity intensity using a smartphone accelerometer. Specifically, the authors compare (1) a single classifier activity-intensity classification of 9 classes with (2) a two-step classification of 3-class activities and 3-class intensities using two separate classifiers. In order to determine the best performing approach for activity-intensity recognition, the authors utilize Analytic Hierarchy Process (AHP) that takes into account the time and memory usage of the different approaches along with their recognition accuracy. Although the overall research framework is adequate, many parts of the manuscript lack clear details. I feel that this manuscript must be re-examined thoroughly in its entirety, and in particular, the following parts must be revised.
1) The details of the dataset must be given. How many data were used for training and testing? How was the feature vector defined (i.e., size etc.)?
In response to the reviewer, we added a mention to the amount of data (in Data Collection subsection) of the new version of the paper. Approximately 10000 samples were collected for each situation.
Regarding the amount used for training and testing, these sizes were defined by the method used, which was 10-folds cross-validation. To explain this better, in the new version of the paper, we added the paragraph below (in Classification subsection), which shortly explains the cross-validation method. We also added a new reference for this method at the end.
“In the test phase, for each of the three mentioned algorithms, the samples were divided into training and test subsets using the k-fold cross-validation method with k = 10. In the k-fold cross-validation, the learning set is partitioned into k disjoint subsets of approximately equal size. This partitioning is performed by randomly sampling cases from the learning set without replacement. The model is trained using k–1 subsets, which, together, represent the training set. Then, the model is applied to the remaining subset, which is denoted as the validation set, and the performance is measured. This procedure is repeated until each of the k subsets has served as the validation set. The average of the k performance measurements on the k validation sets is the cross-validated performance [24].”
2) In section 2.3 (a) the authors describe the process of removing outliers from the collected accelerometer data. Figure 2 shows a dendrogram, but I am not sure what the ‘distance (units)’ and ‘indices’ mean in this context. Does each index signify a data point? Does this mean there are only 30 data points? If the red horizontal dotted line determine the outlier, then based on what? (distance, units?) Do indices #21 and #8 represent the outliers in Figure 2?
The distance is the average Euclidean distance between clusters. We corrected the y-axis label accordingly, and we also corrected the x-axis label, which represents each of the defined clusters. We included an example for better understanding the text, where clusters 8 and 21 represent outliers since they are the clusters most distant from the others (subsection 2.3, item ‘a’):
“a) Removal of corrupted data and outliers. Firstly, all corrupted data (including missing data) returned by the smartphone were removed. Because the data are multivariate, we apply the technique of hierarchical clustering to remove outliers. Hierarchical clustering is an unsupervised technique that calculates the distance between each point present in the database, clustering them according to proximity. To define outliers using this technique, we need to analyze which clusters are divergent from the others based on a threshold of the Euclidean distance. This threshold is drawn in Figure 2 as a horizontal line. We can observe in this figure that clusters 8 and 21 are connected to other ones at much higher distances. Thus we can define both clusters as outliers.”
3) In lines 169-170, the authors state that mean and standard deviation were ‘applied’ (or calculated, perhaps?) to each window. How were these statistics used? It seems that mean and standard deviation were only used for standardizing the data.
Mean and standard deviation were applied both in the feature extraction and standardization steps. To make it clearer, we made corrections to the text to detail the feature vector (subsection 2.3, item ‘b’):
“b) Segmentation and application of statistical metrics. The data were segmented by a window of $2.5$ seconds, without overlap. The following statistical metrics were calculated for each axis (x, y, and z) present in each window: mean and standard deviation. In total, the feature vector had 6 features. After the segmentation process, the dataset contained $2,000$ samples. Metrics were calculated to optimize the data window and to minimize the impact of univariate outliers.”
4) In line 198, the authors observe ‘two’ ‘best’ algorithms (KNN and RF). This expression is strange since ‘best’ usually indicates top ‘one’ of something, not top two.
Thank you for your observation. We corrected the text and showed the changed paragraph in comment #5.
5) I don’t understand what lines 200-201 mean.
The text that generated doubt used the concepts of precision, recall, and accuracy in the area of statistical analysis, for a reading of Table 1. To make the text simpler, we rewrote the paragraph as follows:
“Table 1 shows the results for the first alternative, which uses only one classifier. We can see that KNN and Random Forest obtained better results with 79% and 77% accuracy, respectively. For this reason, these two algorithms were chosen to be compared in more depth later. This same observation can also be made from the value of the F-Score, which takes into account both accuracy and recall.”
6) Table 5 is confusing. The authors compare two different approaches: approach using a single classifier vs. approach using two classifiers. However, the alternative 2 in Table 5 lists three methods, KNN, RF, and RF+KNN. Shouldn’t it list only RF+KNN?
That's correct. There should only be the two best algorithms. We adjusted the table to reflect correctly, as well as the calculation of the AHP analysis, maintaining the criteria but considering only two alternatives.
7) For the two-step approach, the authors report the accuracy to be approximately 80%. I would like to see more detail on how the authors reached this conclusion. Say you have 90% accuracy on activity recognition, and out of the 90% correct data you have 80% correct intensity recognition. In the end, you have 72% of accuracy. In order to reach the conclusion given by the authors, the intensity recognition accuracy given in Table 4 must be completely subsumed by the activity accuracy given in Table 3. Verify if this is the case.
Thank you for pointing out this error. We redid the calculation and updated the text with the new value, as follows.
“In Tables 3 and 4 we conclude that both alternatives obtained similar results for all performance measures evaluated, since it obtained 97% accuracy for activity and 80% for intensity (best scenarios), totaling an accuracy of 78% for the recognition of activity-intensity.”
8) The AHP description given in lines 234-240 is vague and unclear. What does matrix A and matrix B signify? Where do the numbers in equation (2) come from? Detailed explanation is needed.
The text was adjusted to make the reading clearer, as follows.
“The next step is to create the comparison matrix. Thus the matrix is defined based on the expert's knowledge, comparing each attribute according to the importance. For example, in matrix A (column 1: Time, column 2: Memory, and column 3: Accuracy), the experts defined that the accuracy has 7 times the importance of time. Then, to verify the consistency, it is necessary to multiply the comparison matrix A by the priority matrix B. The priority matrix B contains the importance of each criterion. For example, the values shown in Equation 2 show that the most important criterion is accuracy (0.64), followed by memory (0.279), and then by time (0.072).”
9) In lines 251-253, the authors stress the novelty of the activity-intensity aspect. Please refer to the following paper: Pernek et al., “Recognizing the intensity of strength training exercises ith wearable sensors,” Journal of Biomedical Informatics, 58, 2015. (https://www.sciencedirect.com/science/article/pii/S1532046415002142)
Thank you for the suggested bibliographic reference.
A new paragraph has been inserted in the text to address this issue (subsection 2.1 Recognition of Activities Coupled with Intensity):
“According to Pernek et al. [4], the activity considered contains only repetitive weight exercises (six upper body strength training exercises), using five sensors positioned on the arms and chest. The intensity is associated with the muscular strength used in each exercise sequence, using different weight loads. After each series of repetitions, participants were asked to self-assess the intensity of the exercise according to a standard scale. The article considers activity and intensity, but the intensity is interpreted differently from the interpretation that we give in this work, being associated with the strength necessary to execute the movement, and not in the amount of movement.”
10) Last but not least, the manuscript should undergo an extensive proofreading by a native English speaker. Some example of typos and errors are listed below:
- Line 23: activity daily living → missing ‘of’ (activity of daily living)
- Line 56: recognition o distinct
- Line 87: depending of the type → ‘on’ instead of ‘of’
Thank you for pointing out these errors. We have made a careful revision of the English language.
Reviewer 3 Report
The research topic is interesting, and the research problem is significant.
The authors have provided description and some details of the experiments. However, the following comments should be considered to improve the quality of the manuscript.
- The data come from only eight participants. Please comment whether it can be perceived as satisfactory and the data is enough to derive conclusions.
- Please provide more details on the experiments (or probably a separate Methodology section would be useful)
- The idea of „two alternatives” is not clear, e.g. in terms of description in lines 181-185 and 227-229. Also Table 5 is not clear.
- What are the components of A and B matrices? How were they derived?
- What is C matrix?
- Please discuss more on "activity-intensity” approach and the results, as in lines 56-57 it was stated: "The present work focuses on the role of the discrimination of intensity in the recognition o distinct activities”. This statement is neither covered by experiments nor compared with related works.
- In Abstract, lines 10-12 state: "Specifically, the best approach is KNN implemented in the single classifier alternative. The results obtained 97% accuracy for activity recognition and 80% for activity-intensity coupled.”
These results are not consistent with Tables 1-5 (in terms of values or the selected method): in Table 3 we can see that 97% was attained for RF; in Table 1 any of the results equals 80% - Is there any statistical significance of differences in the results?
- Please explain how your findings contribute to the research problem.
- Please verify the writing style of the manuscript and typos, e.g. in Table 5 blank space (0. 77) or Table 6: 0.0.60
Author Response
Reviewer 3:
The research topic is interesting, and the research problem is significant.
The authors have provided description and some details of the experiments. However, the following comments should be considered to improve the quality of the manuscript.
Thank you for your encouraging words and for providing us with constructive comments. All your suggestions have been incorporated into this revision. We hope that you will find the revision satisfactory. In the following, please find our point-by-point responses.
1) The data come from only eight participants. Please comment whether it can be perceived as satisfactory and the data is enough to derive conclusions.
The article already contained the following information in the Data Collection Subsection:
"For the calculation of this sample size, the methodology described in [23] was used, resulting in 8 participants as a function of the small standard deviation in the accelerometer data produced by the device."
We added the following sentence to reinforce the justification for the number of participants in the experiment:
“Other studies have also presented similar experiments with few participants. Pernek et al.[4] assessed how the number of participants impacts the classification of physical exercises, showing that the error of the intensity prediction was stable after adding the sixth participant.”
2) Please provide more details on the experiments (or probably a separate Methodology section would be useful)
We put a new paragraph in the Data Collection subsection providing more details:
“Data collection for each combination of activity and intensity was initiated when the participant pressed a start button. This interaction did not require the smartphone to be removed from the waistband. After 1 min of data collection, a signaling beep was emitted. In the lying posture, the participant used a mat, and in the sitting posture, a chair. In the walking activity, mainly in moderate and vigorous intensities, the participant left the room and walked a 20-meter corridor. The total experiment time for each participant was an average of 15 minutes, which included a training phase of each behavior scenario and the execution of the 9 specific behaviors of 1 min each, with intervals of 30 seconds.”
3) The idea of „two alternatives” is not clear, e.g. in terms of description in lines 181-185 and 227-229. Also Table 5 is not clear.
Concerning lines 181-185, we rewrote the paragraph to make the understanding clearer, as follows:
“We defined two possible “classification alternatives” for Activity-Intensity recognition. The first alternative is to classify the data directly using only one classifier. In this case, we will have a classifier that infers from the 9 possible classes, for example, light walking. In the second alternative, a first classifier infers the activity, for example, walking, and a second classifier infers the intensity, for example, light. The second alternative would be important to mitigate the inter-class variability, since we will first recognize the activity, probably with high accuracy, and then the intensity, which classification is more difficult."
For lines 227-229 we rewrite two paragraphs:
“In Tables 3 and 4, we conclude that both alternatives obtained similar results for all performance measures evaluated, since it obtained 97% accuracy for activity and 80% for intensity (best scenarios), totaling an accuracy of 78% for the recognition of activity-intensity. In this alternative, when obtaining the activity with the first classifier (e.g. walking), the second classifier will only define the intensity of the activity (light, moderate, and vigorous) already defined. This probably improves the efficiency of the second stage of classification.”
“Using the second alternative, with two classifiers placed in sequence, the combination of algorithms with the best results were RF (activity) + KNN (intensity), with 78% of total accuracy. However, the recognition of activities is performed through an application on a smartphone, which has a shortage of resources, especially the battery. Therefore, the chosen classifier must have the lowest cost in terms of consumption of device resources, and with the greatest accuracy.”
In respect to Table 5, there should only be the two best algorithms. We adjusted the table to reflect correctly, as well as the calculation of the AHP analysis, maintaining the criteria but considering only two alternatives.
4) What are the components of A and B matrices? How were they derived?
The text was adjusted to make the reading clearer.
“The next step is to create the comparison matrix. Thus the matrix is defined based on the expert's knowledge, comparing each attribute according to the importance. For example, in matrix A (column 1: Time, column 2: Memory, and column 3: Accuracy), the experts defined that the accuracy has 7 times the importance of time. Then, to verify the consistency, it is necessary to multiply the comparison matrix A by the priority matrix B. The priority matrix B contains the importance of each criterion. For example, the values in Equation 2 show that the most important criterion is accuracy (0.64), followed by memory (0.279), and then by time (0.072).”
5) What is C matrix?
Matrix C is the result of multiplying matrices A and B. It was already in the text in Equation 3, but the name was missing. The text has been corrected.
6) Please discuss more on "activity-intensity” approach and the results, as in lines 56-57 it was stated: "The present work focuses on the role of the discrimination of intensity in the recognition o distinct activities”. This statement is neither covered by experiments nor compared with related works.
The word "discriminating" in the aforementioned sentence was used to separate, separate the intensity of the activity so that both could be assessed individually. So this statement was covered by the article, as it was a premise of the work. But to improve the text, we added the paragraph below in the Conclusion.
“In this work, we investigate the use of machine learning techniques to recognize human activities and postures at different intensities. The coupling of the activity and intensity parameters allowed the recognition of the human activities considered here (walking, sitting, and lying down) in a more precise and complete way. The results obtained 97% accuracy for activity recognition and 78% for activity-intensity coupled. The automatic identification of movements/postures is feasible and useful in the context of health care, where the intensity of movements linked to each specific posture or activity has a distinct clinical repercussion. For example, vigorous movements while lying down can be correlated with sleep disorders. At the same time, intense movements during walking are not necessarily indicative of illness. Thus, the automated provision of this information can assist in the decision making of health professionals.”
7) In Abstract, lines 10-12 state: "Specifically, the best approach is KNN implemented in the single classifier alternative. The results obtained 97% accuracy for activity recognition and 80% for activity-intensity coupled.”
These results are not consistent with Tables 1-5 (in terms of values or the selected method): in Table 3 we can see that 97% was attained for RF; in Table 1 any of the results equals 80%
Thank you for pointing out this error. The error arose because of confusion between the two alternatives. We corrected the Abstract to solve this, as follows:
“The best approach was KNN implemented in the single classifier alternative, which resulted in 79\% of accuracy. Using two classifiers, the result was 97% accuracy for activity recognition (Random Forest), 80% for intensity recognition (KNN), which resulted in 78% for activity-intensity coupled.”
8) Is there any statistical significance of differences in the results?
We are not interested in generalizing the result at this time, as we are interested in using Machine Learning assessment metrics only to obtain a suggestion of the best method, without an evaluation of statistical significance. In future works, we intend to perform a chi-square test to verify if the difference of accuracy between the KNN, RF, and KNN+RF classifier methods is statistically significant.
9) Please explain how your findings contribute to the research problem.
This issue is addressed in comment #6.
10) Please verify the writing style of the manuscript and typos, e.g. in Table 5 blank space (0. 77) or Table 6: 0.0.60
All typos were corrected in the tables cited and throughout the paper, after a careful proofreading.
Round 2
Reviewer 2 Report
I believe that the manuscript has been significantly improved. To further improve the manuscript, I advise the following aspects to be addressed.
- Insert an equal sign between the letter A and the matrix in equation (2).
- Describe how the values in Table 6 was calculated; that is, explain how the values ‘0.86’ and ‘0.58’ were calculated using the weights given by the matrix B.
Author Response
Thank you for your comments, which helped to improve the quality of the manuscript.
The performance values in Table 5 are time, memory consumption, and accuracy. Time and memory are "the smaller the better" values. To standardize all values as performance, we calculate the inverse (1 / x) for time and memory. Accuracy is maintained at its original values, as it is already a performance measure of the type "the bigger the better".
So the table is:
22.22222222 6.896551724 0.79
4.255519199 1.569858713 0.78
To normalize these values, we divide each value by the sum of the column (except accuracy):
0.839285714 0.814578005 0.79
0,160714286 0,185421995 0,78
The final measure of the AHP score is the weighted average of the normalized performance parameters, using the values of matrix B as weights.
This results in:
0.84x0.072 + 0.81x0.279 + 0.79x0.649 = 0.80
0.16x0.072 + 0.19x0.279 + 0.78x0.649 = 0.57
After this double-check, we noticed some rounding errors.
We changed the text of the article to better explain this calculation methodology, and to make the results reproducible, as follows:
"To calculate the AHP scores, we need to standardize all performance values. We calculate the inverse (1 / x) for time and memory because these are inversely proportional to the accuracy. Accuracy is maintained at its original values, as it is already a performance measure of the type "the bigger the better". Then, each value is normalized by the sum of each column. The best algorithm is defined by the weighted average of the normalized performance parameters, using the values of Matrix B as weights. Table 6 shows the result. "
Reviewer 3 Report
The authors responded to all my comments. I accept the paper in the present form.
Author Response
Thank you for your comments, which helped to improve the quality of the manuscript.
This manuscript is a resubmission of an earlier submission. The following is a list of the peer review reports and author responses from that submission.